# Ionotropic Glutamate Receptors in Epilepsy: A Review Focusing on AMPA and NMDA Receptors

**DOI:** 10.3390/biom10030464

**Published:** 2020-03-18

**Authors:** Takahisa Hanada

**Affiliations:** Medicine Development Center, Eisai Co., Ltd., Koishikawa 4-6-10, Bunkyo-ku, Tokyo 112-8088, Japan; t-hanada@hhc.eisai.co.jp; Tel.: +81-3-3817-5149

**Keywords:** AMPA, epilepsy, glutamate, ictogenesis, NMDA, pharmacology

## Abstract

It is widely accepted that glutamate-mediated neuronal hyperexcitation plays a causative role in eliciting seizures. Among glutamate receptors, the roles of N-methyl-D-aspartate (NMDA) and α-amino-3-hydroxy-5-methylisoxazole-4-propionic acid (AMPA) receptors in physiological and pathological conditions represent major clinical research targets. It is well known that agonists of NMDA or AMPA receptors can elicit seizures in animal or human subjects, while antagonists have been shown to inhibit seizures in animal models, suggesting a potential role for NMDA and AMPA receptor antagonists in anti-seizure drug development. Several such drugs have been evaluated in clinical studies; however, the majority, mainly NMDA-receptor antagonists, failed to demonstrate adequate efficacy and safety for therapeutic use, and only an AMPA-receptor antagonist, perampanel, has been approved for the treatment of some forms of epilepsy. These results suggest that a misunderstanding of the role of each glutamate receptor in the ictogenic process may underlie the failure of these drugs to demonstrate clinical efficacy and safety. Accumulating knowledge of both NMDA and AMPA receptors, including pathological gene mutations, roles in autoimmune epilepsy, and evidence from drug-discovery research and pharmacological studies, may provide valuable information enabling the roles of both receptors in ictogenesis to be reconsidered. This review aimed to integrate information from several studies in order to further elucidate the specific roles of NMDA and AMPA receptors in epilepsy.

## 1. Introduction

The central nervous system utilizes various substances as neurotransmitters. Among them, glutamate and gamma aminobutyric acid (GABA) are the major neurotransmitters for excitatory and inhibitory function, respectively. In the cerebral cortex, approximately 70%–80% of neurons are glutamatergic neurons, with the remainder comprising GABAergic interneurons [1,2]. Thus, it is evident that glutamatergic and GABAergic neurons primarily compose basic neuronal networks, especially in the cortex. Cortical pyramidal neurons possess approximately 30,000 synapses, of which 95% are excitatory synapses [3], indicating that glutamate is the principal excitatory neurotransmitter in the brain, and also that GABAergic inhibition influences neuronal activity in an efficient manner [4]. Recently, it has been demonstrated that glutamate is also utilized as a gliotransmitter, released from glial vesicles and channels, with evidence suggesting that glutamate released from glial cells modulates synaptic efficiency [5,6] and controls the release of various biological molecules, including cytokines [7]. Therefore, it is possible that glutamate may contribute to various physiological/pathological conditions. Control of glutamatergic neuronal activity seems, therefore, the most important paradigm for maintaining normal brain activity. An imbalance in neuronal excitation and inhibition, which could be caused by a variety of possible changes occurring within a neuronal network, is an established and well-accepted hypothesis for the pathogenesis of epilepsy [8]. This hypothesis describes a major role for glutamate at excitatory synapses in transmitting an excitatory signal to other neurons. An increase in glutamate levels in the extracellular fluid during the ictal phase (i.e., during a seizure) is well documented [9,10,11], and glutamate receptor agonists such as domoate are known to elicit seizures in humans and animals [12,13,14], while N-methyl-D-aspartate (NMDA), α-amino-3-hydroxy-5-methylisoxazole-4-propionic acid (AMPA), and kainate have been reported to elicit seizures in rodent models [15,16]. These reports indicate that glutamate receptors, particularly ionotropic glutamate receptors, have a significant role in ictogenesis (i.e., the generation of a seizure).

Extracellular glutamate concentration is strictly controlled in the central nervous system due to its neurotoxic effects at high concentrations [17]. Glutamate transporters, autoreceptors, and desensitization of postsynaptic receptors all contribute to the control of glutamatergic signals [17,18]. As noted, direct activation of glutamate receptors can elicit seizures; however, the pathological function of each glutamate receptor in epilepsy is not well elucidated. Glutamate receptors comprise two large subclasses: ionotropic glutamate receptors and metabotropic glutamate receptors. Ionotropic glutamate receptors are ligand-gated ion channels, whereas metabotropic glutamate receptors are G-protein-coupled receptors utilized as second messengers [19,20]. Among ionotropic glutamate receptors, the NMDA-type and AMPA-type glutamate receptors are the most well studied in terms of their physiological roles. Antagonists of these receptor types have also been evaluated as anti-seizure drugs in clinical trials. The NMDA antagonists, CPP-ene and MK-801, failed to demonstrate efficacy and produced severe psychotomimetic adverse events in subjects with focal seizures, which were not observed in healthy volunteers [21,22]. Remacemide, which is a weak inhibitor of NMDA receptors, demonstrated efficacy as adjunctive therapy in patients with focal seizures; however, as monotherapy, remacemide was clearly inferior to carbamazepine in patients with focal seizures [23,24]. On the other hand, an AMPA antagonist, perampanel, has been approved as an anti-seizure drug for focal seizures and primary generalized tonic-clonic seizures [25,26,27,28,29]. Recent preliminary observations have suggested that perampanel may be efficacious in patients with other seizure types such as myoclonic or absence seizures [30,31]. As noted above, agonists of both NMDA and AMPA receptors have been shown to elicit seizures in animal models, but clinical study results differ between the two receptor types. This inconsistency might be due to a limited understanding of each receptor’s function in epilepsy, and this review aims to clarify the role of these two major glutamate receptor types in epilepsy from various points of view.

## 2. AMPA- and NMDA-type Glutamate Receptors

### 2.1. AMPA Receptors

AMPA receptors are directly activated by the binding of glutamate; no other endogenous ligand has been discovered to date. They are predominantly expressed in the postsynaptic neuronal membrane and play a role in rapid excitatory neurotransmission in the brain network. AMPA receptors are tetramers composed of four types of subunit (GluA1 to GluA4), and can be either homo- or hetero-tetrameric. The GluA2 subunit is the determinant of calcium permeability; GluA2-containing calcium-impermeable AMPA receptors are mainly expressed in excitatory projection neurons [19], whereas calcium-permeable AMPA receptors (lacking the GluA2 subunit) are mainly expressed in inhibitory interneurons [32,33]. Expression of calcium-permeable AMPA receptors has also been observed at excitatory neuronal synapses under conditions of synaptic plasticity, including physiological plasticity and after pathological insult [7,34,35]. AMPA receptor turnover at synapses is considered continuous and fast. Synaptic potentiation and depression are regulated by alterations in the number of synaptic AMPA receptors, and the machinery and mechanisms involved in the trafficking of AMPA receptors into and out of the synapse are now the focus of intensive research [36]. This is because changes in synaptic plasticity mediated by receptor trafficking could contribute to pathological alterations in brain network signaling.

AMPA receptor expression is not restricted to the neurons; all glial cell types express AMPA receptors [36]. Expression levels in astrocytes can vary according to circumstances including inflammation [37]. Microglia and macrophages also express AMPA receptors, with AMPA receptor activation contributing to the release of pro-inflammatory cytokines [36,38]. Additionally, AMPA receptors are expressed in inflammatory/immune cells [39], where they have a role in proliferation, cell adhesion, chemotaxis, and release of pro-inflammatory cytokines [40,41,42]. This relatively varied distribution suggests that the AMPA receptor may have various physiological and pathological roles.

### 2.2. NMDA Receptor

NMDA receptors mainly exist in the postsynaptic membrane, but are also expressed in pre-synaptic membranes. Functional NMDA receptors comprise two GluN1 subunits together with either two GluN2 subunits or a combination of GluN2 and GluN3 subunits, and require either glycine or D-serine to act as a co-agonist for activation [43]. Quinolinic acid and kynurenic acid are also known to be an endogenous agonist and antagonist, respectively, of NMDA receptors [44], and receptor activity is modulated by protons and Zinc. The NMDA receptor channel is blocked by Mg^2+^ at neuronal resting membrane potential, and Mg^2+^ is removed when the membrane is depolarized. NMDA channel kinetics are much slower compared with the AMPA receptor, and NMDA receptor desensitization kinetics differ according to GluN2 subunit composition. In synaptic transmission, the activation of NMDA receptors is slow and prolonged due to the requirement for membrane depolarization in addition to the slower activation and desensitization kinetics [43]. Thus, the NMDA receptor channel possesses complex machinery controlling receptor activity, suggesting that it may have a different physiological function to the AMPA receptor. The distribution of NMDA receptors is not limited to neurons; astrocytes also express NMDA receptors, however their function in this setting is not well elucidated [45].

## 3. Pharmacological Induction of Seizures by AMPA- and NMDA-Receptor Manipulation

Pharmacological experiments have clearly indicated that agonists of ionotropic glutamate receptors act as convulsants. For example, infusion of AMPA, kainate, (RS)-2-amino-3-(3-hydroxy-5-tert-butylisoxazol-4-yl)propanoic acid (ATPA), and NMDA has been shown to elicit clonus and tonus in rodents [15,16], indicating that both NMDA- and AMPA-type glutamate receptors contribute to ictogenesis. Kainate, which is an agonist of the kainate receptor and also acts as a non-desensitized agonist of the AMPA receptor [46], is often utilized as an animal model of temporal lobe epilepsy and status epilepticus [47]. As noted above, intoxication caused by the marine toxin domoic acid, a kainic acid analog, has been reported in humans. Intoxicated patients experienced drug-resistant status epilepticus and developed temporal lobe epilepsy within one year, such that the consequences of human domoic acid intoxication were very similar to the rodent model of kainate-induced status epilepticus. These results may suggest that over-activation of AMPA receptors could elicit temporal lobe epilepsy, which could be explained by the relatively dense expression of the AMPA receptor in the hippocampus [48,49]. On the other hand, NMDA-induced seizures in adult rodents are not well characterized, and the corresponding human seizure type is unclear. In infant rodents, injection of NMDA has been reported to elicit a spasm-like phenotype, which has been utilized for drug evaluation [50,51] and may be attributable to the slower kinetics of the NMDA receptor compared with the AMPA receptor.

The rat amygdala kindling model is a widely used model of temporal lobe epilepsy and has been used to evaluate the effects of pharmacological manipulation of AMPA and NMDA receptors on seizure induction and severity. Generally, AMPA receptor antagonists have shown efficacy in this animal model [52,53,54,55], whereas competitive and non-competitive NMDA antagonists have only demonstrated weak efficacy [53,56]. However, Rundfeldt et al. demonstrated that administration of an NMDA glycine-site antagonist (7-CKA; 7-chlorokynurenic acid), a low-efficacy partial agonist ((+)-HA-966; R(+)-3-amino-l-hydroxypyrrolid-2-one), and a high-efficacy partial agonist (D-CS; D-cycloserine) exerted potent anticonvulsant effects in fully kindled rats, with all three drugs increasing the after-discharge threshold (ADT; i.e., the level of electrical stimulation required to induce a seizure) for focal seizures. The most potent anticonvulsant effects were observed with the glycine site antagonist 7-KCA, followed by (+)-HA-966 and D-CS [57]. In another study, Potschka et al. demonstrated that the glutamate receptor antagonist, LU 73,068 (4,5-dihydro-1-methyl-4-oxo-7-trifluoromethyl-imidazo [1,2*a*]quinoxaline-2-carbonic acid), which binds with high affinity to both AMPA receptors and the glycine site of NMDA receptors, dose-dependently increased the ADT for focal seizures, and completely blocked seizures when rats were stimulated with a current 20% greater than the prespecified control ADT [58]. Additionally, combined administration of a non-effective dose of the non-NMDA receptor antagonist NBQX (2,3-dihydroxy-6-nitro-7-sulphamoyl-benzo[f]quinoxaline), which binds with high affinity to AMPA receptors, and the NMDA glycine-site antagonist L-701,324 (7-chloro-4-hydroxy-3-(3-phenoxy)phenyl-quinoline-2(1H)one) elicited a significant increase in ADT, whereas neither drug exerted any anticonvulsive effects when administered alone, indicating a synergistic interaction and suggesting that AMPA and NMDA receptors may play different roles in ictogenesis. Furthermore, Wlaź et al. reported that acute administration of D-CS dose- and time-dependently increased the threshold for tonic seizures in Naval Medical Research Institute mice at doses that did not induce motor impairments, whereas the uncompetitive NMDA receptor antagonist MK-801 (dizocilpine) increased the seizure threshold but also elicited motor impairments. Interestingly, combined administration of D-CS with MK-801 potentiated the MK-801–induced motor impairments but not the anticonvulsive effects, which were merely additive, suggesting that the NMDA glycine binding site differentially modulates distinct pharmacodynamic actions of NMDA receptors [59]. These results suggest that the complexity of NMDA-receptor activation may affect anti-ictogenic responses to drugs which are antagonists for the different modulatory sites of NMDA receptors.

Taken together, these results may suggest that these different types of ionotropic glutamate receptors, and their different modulatory sites, could have different roles in ictogenesis, and that an increased understanding of their roles and function may aid the development of new anti-seizure drugs.

## 4. NMDA- and AMPA-Receptor–Related Epilepsy in Humans

### 4.1. Anti-NMDA Antibody Encephalitis

Recently, autoimmune encephalitis has been reported to be an entity of epilepsy [60]. Anti-NMDA receptor encephalitis is a major type of autoimmune encephalitis; anti-NMDA receptor antibodies cause internalization of surface NMDA receptors, resulting in a decrease in receptor density on the cellular surface [61,62,63,64]. Previous studies in rodents provided no evidence that NMDA antagonists elicit seizures [65,66], and anti-NMDA encephalitis seems controversial in the light of positive results with NMDA antagonists in rodent studies [67,68,69,70]. However, it has been documented that D-CPP-ene, a competitive NMDA antagonist, worsened seizures in 3/8 patients with epilepsy in a clinical study [21], suggesting that acute reduction of NMDA receptor function may result in an excitatory and inhibitory imbalance. Additionally, an autoantibody purified from patients with NMDA encephalitis increased extracellular glutamate levels in the rodent brain [65]; it was postulated that the autoantibody caused GABAergic dysregulation, since GABAergic interneurons utilize NMDA receptors for neuronal excitation [65]. Similarly, it is accepted that ketamine inhibits GABAergic interneuron activity and causes a surge of extracellular glutamate after administration in animals and humans [71,72]. Disinhibition of excitatory neurons could also potentially explain the cause of seizure in anti-NMDA encephalitis, such that both the activation of NMDA receptors in glutamatergic neurons and the inhibition of NMDA receptors in GABAergic interneurons could result in the occurrence of seizures. This may explain to some extent the conflicting results among different conditions.

### 4.2. Genetic Mutations in the NMDA Receptor

The genetic mutations in NMDA receptors that may cause epilepsy in humans were summarized by Xu et al. [73]. Mutations in *GRIN1* (which encodes the GluN1 subunit), *GRIN2B* (GluN2B), and *GRIN2D* (GluN2D), expressed during embryonic development, display more severe clinical phenotypes, including severe intellectual disability and developmental delay, than *GRIN2A* (GluN2A) mutations. In addition, more than half of GluN1 mutations are loss-of-function mutations. GluN1 is the essential subunit for a functional NMDA receptor, suggesting that mutations in *GRIN1* would exert a significant impact on neuronal activity [43]. Interestingly, *GRIN1* mutation seizure phenotypes exhibit variable semiology (spasms, tonic and atonic seizures, hypermotor seizures, focal dyscognitive seizures, febrile seizures, generalized seizures, status epilepticus, myoclonic seizures, etc.) and electroencephalogram (EEG) patterns (hypsarrhythmia, focal, multifocal and generalized spikes and waves), and appear to be independent of channel function (both loss-of-function or gain-of-function *GRIN1* mutation phenotypes exhibit seizures) [74,75].

The seizure types most commonly observed in patients with GluN2A mutations, including both loss-of-function and gain-of-function mutations, are benign epilepsy with centro-temporal spikes (BECT), atypical benign partial epilepsy, continuous spike and wave during slow-wave sleep (CSWS), and Landau–Kleffner syndrome (LKS); some patients also display motor and language disorders [76,77,78,79,80]. However, a de novo gain-of-function mutation with a clinical presentation that could not be defined by a specific epileptic syndrome has also been reported [81].

With regard to *GRIN2B*, gain-of-function mutations result in West syndrome and other seizures [82,83].

Functional changes of mutated NMDA receptor subunits can be categorized as loss of function, gain of function, or no change in function; loss-of-function mutations are the most commonly observed. As noted, the GluN1 subunit is essential for a functional NMDA receptor, suggesting that a loss-of-function mutation may produce a similar phenotype to anti-NMDA encephalitis. Differences in clinical presentations could be explained temporally; NMDA receptor encephalitis is an acutely acquired condition, whereas *GRIN1* encephalopathy resulting from a loss-of-function mutation represents a chronic neurodevelopmental disease. However, a number of symptoms, including choreatic and dystonic movements, seizures, and sleep-cycle dysregulation, can be observed in both conditions, indicating that similarity exists between hypo-NMDA-receptor-function–related diseases.

Gain-of-function mutations in *GRIN1* directly cause overexcitation of NMDA receptors, and, in addition to gain-of-function mutations in other genes related to increased NMDA-receptor function, are classified as causing ‘NMDA-pathy’ [84]. These mutations cause epileptic spasms and tonic, focal, myoclonic, local migrating, or altering seizures, with the following EEG phenotypes: suppression burst, multifocal spikes, hypsarrhythmia, slow spike waves, and CSWS. Physiologically, the NMDA receptor produces slower and longer excitation compared with the AMPA receptor; the seizure types and EEG phenotypes produced by NMDA receptor gain of function would therefore suggest that longer abnormal excitation plays a role in producing these disease phenotypes.

The existence of both hypo-NMDA-receptor function and enhanced NMDA-receptor function across disease phenotypes suggests that NMDA-receptor–related epilepsy cannot be simply explained. Comparison of receptor function between mutated NMDA receptor phenotypes and anti-NMDA encephalitis suggests two potential pathological pathways: “hypo-NMDA function” and “hyper-NMDA function”. Hypo-NMDA function produces a severe phenotype, including hyperkinesia, epilepsy, and cognitive impairment, while hyper-NMDA function produces various seizure types and is often associated with prolonged electrical activity. As demonstrated in Figure 1, both hypo- and hyper-NMDA function produce excitatory overstimulation. This can be explained in part by the fact that GABAergic neurons and inhibitory synapses are far fewer in number relative to glutamatergic neurons and excitatory synapses [1,2,3,71,72], such that a state of reduced excitability (hypo-NMDA function) resulting in increased GABAergic neuronal inhibition is unlikely. Additionally, excitatory over-stimulation due to hyper-NMDA function could therefore easily outweigh GABAergic inhibition, again resulting in enhanced neuronal excitation.

### 4.3. Genetic Mutations in the AMPA Receptor

Mutations in the AMPA receptor are not as commonly reported compared with the NMDA receptor. AMPA receptor gene mutations are often associated with cognitive impairment and autism spectrum disorders, and sometimes with epilepsy [85,86,87,88]. Recently, Salpietro et al. [89] reported that 28 unrelated individuals presenting with neurodevelopmental abnormalities and seizures or developmental epileptic encephalopathy had heterozygous de novo *GRIA2* mutations. Functional analyses revealed loss of function for the majority of the mutations, and a number of mutated receptors showed inward rectification, suggesting a change of channel activity. Disease severity did not correspond to functional changes in receptors. Some of the mutations identified resulted in alterations in the levels of GluA2-containing surface receptors, suggesting that these mutations may cause defects in surface trafficking or heteromerization, thereby altering receptor subunit composition, which could in turn impact on cellular excitability. The seizure types observed in the patients in this study were focal, tonic-clonic, clonic, and tonic, suggesting that the phenotypes induced by AMPA receptor mutations cannot be considered simply. Further research in transgenic animal models is needed to increase understanding of AMPA receptor mutation phenotypes.

Evidence for a direct link between AMPA receptor mutation and epilepsy is limited, whereas cognitive impairment and autism appear to be established phenotypes. It is possible that developmental changes in the brain resulting from AMPA receptor mutation may contribute to the presentation of an epileptic phenotype.

Turnover of AMPA receptors at the synapse is relatively rapid, meaning that a number of newly synthesized receptors contribute to a change in synaptic potential in response to a stimulus, leading to plastic change. The physiological machinery of receptor trafficking, under both physiological and pathological conditions, represents a major focus in the study of synaptic plasticity [34,35,36]. Increased expression of AMPA receptors in the brain has often been described or suggested in various types of epilepsy [90,91,92], whereas a similar trend in the density of NMDA receptors has not been observed [90,91,93,94,95]. Additionally, expression levels of certain AMPA and NMDA receptor subunits have been reported to increase in the human epileptic brain, suggesting alterations in receptor function [96,97,98]. This increased receptor expression and alteration of subunit composition may be related to cortical hyperexcitability in the epileptic brain.

AMPA receptor accumulation is frequently found in dissected brain tissue from patients with focal seizures. The causes of accumulation have not yet been fully elucidated; however, multiple reports have demonstrated a relationship between deficits in machinery of AMPA-receptor retrieval and seizure-related disorders.

Thorase, an ‘ATPase Associated with diverse cellular Activities’ (also referred to as an AAA+ ATPase) encoded by *ATAD1*, was discovered by screening for neuroprotective genes against excitotoxic insults [99,100]. Thorase regulates surface AMPA receptor expression through its mediation of the disassembly of the GluA2–GRIP complex [101]. A homozygous mutation in *ATAD1* was found in three siblings who presented with a severe, lethal encephalopathy associated with a severe neurologic disorder characterized by hypertonia and seizures. Additionally, an animal model with genetic deletion of Thorase showed substantial reduction in AMPA receptor internalization, leading to increased amplitudes of miniature excitatory postsynaptic currents, enhancement of long-term potentiation, and elimination of long-term depression. Homozygote Thorase knockout mice are viable, however approximately 80% die of a seizure-like syndrome [102]. These findings in transgenic animals are reproduced in the phenotypes of human cases; treatment with perampanel in both human cases and mice improved seizure-related symptoms and slowed neurodegeneration [103].

At least three different missense mutations in the *Nedd4-2* gene have been identified in idiopathic generalized epilepsy with photosensitivity [104,105,106]. Nedd4-2 is an E3 ubiquitin ligase, and these mutations disrupt Nedd4-2 binding to 14-3-3+, thereby reducing its capacity for ubiquitination of GluA1, resulting in increased surface expression of AMPA receptors. This explains the apparent elevation in seizure susceptibility in Nedd4-2–deficient transgenic mice [107]. Wu et al. [108] demonstrated that expression of Nedd4-2 is downregulated in a rat model of mesial temporal lobe epilepsy (MTLE). The decreased expression of Nedd4-2 was closely associated with spontaneous seizures in the late phase, which corresponds to the timing of increased AMPA receptor expression in a human MTLE sample. The authors concluded that inhibition of ubiquitin-proteasome system may aggravate epileptogenesis, and that Nedd4-2 is a critical E3 ubiquitin ligase involved in this process.

Mutations in the *RAB39B* gene are reported to cause intellectual disability comorbid with autism spectrum disorder and epilepsy [109]. Mignogna et al. [110] described the interaction of RAB39B with its downstream effector PICK1 as a key component of GluA2 AMPA receptor subunit trafficking. RAB39B dysfunction causes impaired trafficking of GluA2/GluA3 to the Golgi compartment, resulting in increased levels of non–GluA2-containing calcium-permeable forms of the AMPA receptor. AMPA receptors lacking the GluA2 subunit are predominantly observed in young animals or after an event of plasticity-inducing neuronal activity [7,33,34,35], while mouse *GRIA2* knockouts, lacking GluA2, exhibit a greater magnitude of long-term potentiation and non-saturated long-term potentiation [111]. This transgenic animal observation might suggest that the GluA2-lacking brain may enhance its activity by repetitive, strong excitation. Furthermore, RAB39B mutation and stress-induced calcium-permeable AMPA receptor expression may result in an increase in unstable synapses, leading to the emergence of a dysregulated, hyperexcitable network.

These three receptor-trafficking–related gene mutations could cause hyperexcitability elicited by changes in cell-surface receptor expression. Abnormal AMPA receptor expression on the cell surface is also known to occur in diseases involving intracellular substance accumulation. The AMPA antagonist perampanel is now recognized as a potentially efficacious drug for progressive myoclonic epilepsies (PME) [112,113,114,115,116,117,118,119,120,121,122,123], a group of rare types of epilepsy, most of which are recognized as intracellular substance storage disorders. Although each of these diseases involves increased intracellular storage of a different pathological substance, their clinical phenotypes are very similar; it is therefore reasonable to consider the existence of several similar AMPA-receptor–related endophenotypes that produce the same clinical phenotype. Perampanel has demonstrated efficacy in a broad range of PME, including Unverricht–Lundborg disease, Lafora disease, DRPLA (dentatorubral-pallidoluysian atrophy), MELAS (mitochondrial encephalomyopathy, lactic acidosis, and stroke-like episodes), and ceroid lipofuscinosis. The AMPA-receptor–mediated response in some animal models of PME has also been evaluated. Malin knockout mice, a model of Lafora disease, showed increased synaptic potentiation after high-frequency stimulation [124]. Given that an increase or decrease in synaptic transmission following high-frequency stimulation is caused by an increase or decrease, respectively, of postsynaptic AMPA receptor expression [125], this effect in Malin knockouts is indicative of increased postsynaptic accumulation of AMPA receptors. Similarly, neurons from a transgenic animal model of juvenile Batten disease (also called CLN3 disease) have been shown to be susceptible to excitotoxicity [126]. Kovács et al. investigated the cause of this increased glutamate sensitivity by evaluating AMPA receptor expression at the neuronal surface in the CLN3 mouse model, and reported increased surface expression of AMPA receptors in various areas of the brain [127]; furthermore, the behavioral phenotype of the CLN3 model was ameliorated by the AMPA antagonist EGIS-8332 [128]. Additionally, in an electrophysiological study of brain slices from a mouse model of Niemann-Pick type C disease, D’Arcangelo et al. demonstrated impairment of AMPA-induced receptor internalization [129], and, correspondingly, another study reported increased surface expression of GluA2-containing AMPA receptors in neurons derived from Niemann-Pick type C1 patient-specific induced pluripotent stem cells [130].

Reports on mutations in Thorase and *Nedd4-2* and receptor expression in substance storage disorders have indicated that impairment in receptor trafficking, particularly in retrieval from the cell surface, is the probable cause of the epileptic phenotype. With the exception of *Nedd4-2* mutation-associated disorders, diseases in which impairment in receptor trafficking is observed are rare and serious disorders. Impairment of the receptor-trafficking system may not have a causative role across more common seizure disorders such as focal seizures; however, evaluation of surgically dissected brain samples from patients with temporal lobe epilepsy indicated an increase in AMPA receptor density. Additionally, electrophysiological analysis of brain slice specimens from patients with interictal-spike–like electrical activity showed further increases in AMPA receptor density compared with slices without interictal-spike–like activity [91]. Interestingly, GABA receptor density is not altered or reduced, and NMDA receptor density did not show consistent changes, in patients with focal seizures [90,91,93,94,95]. These autoradiograph studies suggest that the AMPA receptor is involved in hyperexcitability, at least in the ictogenic zone, and that there is a functional alteration of the AMPA receptor in the human epileptic brain. The electrophysiological characteristics and ion permeability of AMPA receptors is determined by alternative splicing (flip/flop) and RNA editing (at the R/G and Q/R sites of GluA2) [131,132,133]. Surgically dissected brain samples from patients with temporal lobe epilepsy showed a significant increase in the relative amount of edited RNA at the GluA2 R/G site in hippocampal tissue, but not in the cortex, compared with autopsy-derived control brain tissue. AMPA receptor isomers containing Glycine at the GluA2 R/G site have a faster recovery rate from desensitization compared with Arginine-containing isoforms [134]. As such, increased RNA editing at the R/G site could result in an enhanced response to glutamate in the epileptic brain due to faster recovery from desensitization.

As previously described, animal models of kainate-induced seizures suggest that excessive whole-brain activation of AMPA receptors elicits temporal lobe seizures, indicating higher AMPA receptor expression levels in the hippocampus. Excessive brain excitability due to increased AMPA receptor expression at the seizure focus could be a cause of focal seizures in humans. Mutations in *ATAD1* and *Nedd4-2* produce upregulated AMPA receptor expression at the neuronal surface, while increased cell surface expression of AMPA receptors may also underly generalized seizure disorders, particularly substance storage disorders. Combined, the evidence described here indicates that increased AMPA receptor expression may represent a common endophenotype among seizure disorders, and may support the utility of AMPA antagonists as broad-spectrum anti-seizure drugs (Figure 2).

## 5. Role of NMDA and AMPA Receptors in Ictogenesis

The knowledge accumulated to date supports a possible contribution of AMPA and NMDA receptors to the pathophysiology of epilepsy, however their role in ictogenesis remains unclear. An ictal event is defined by hypersynchronized electrical activity, and results from transcranial magnetic stimulation studies demonstrate that the epileptic brain shows hyperexcitability in the interictal phase [135,136], with direct current (DC) shifts [137] and high-frequency oscillations observed just prior to the ictal event [137,138,139,140]. Therefore, ictogenesis in focal seizures can be divided into several steps: hyperexcitability in the interictal phase, DC shift/high-frequency oscillation, and hypersynchronization. These individual steps could represent targets for pharmacological manipulation by anti-seizure drugs. The contribution of AMPA and NMDA receptors to each step of ictogenesis is summarized below.

### 5.1. Background Irritability

An imbalance between neuronal excitation and inhibition is generally accepted to be the cause of epilepsy. Various factors can contribute to the imbalance, typically insufficiency of GABAergic inhibition, loss of glutamate clearance systems, ectopic release of glutamate from non-neuronal cells, etc. Various methods have been employed to measure glutamate levels in patients with epilepsy. For example, Stover et al. demonstrated an elevation of glutamate levels in cerebrospinal fluid (CSF) from epilepsy patients, and interestingly, the CSF glutamate levels recorded were higher than those observed in stroke patients [141]. Additionally, During et al. [10] reported that glutamate levels measured using micro-dialysis were elevated before the initiation of ictal activity, whereas GABA levels were not, leading the authors to suggest that an elevation of glutamate prior to an ictal event may be attributable to non-neuronal cells. A separate study also demonstrated that glutamate levels around the epileptogenic zone during the interictal period are higher than in non-epileptogenic zones, with the highest glutamate level observed at the exact seizure focus [142]. Glutamate chemical exchange saturation transfer (GluCEST) is a technique for evaluating brain glutamate levels using 7 Tesla magnetic resonance imaging. Evaluation in epilepsy patients indicated a higher GluCEST signal in the hippocampus ipsilateral to the location of seizure onset, again suggesting an increase in glutamate levels during the interictal phase [143]. Furthermore, increased peritumoral GluCEST contrast in glioma patients was associated with both recent seizures and drug-refractory epilepsy [144]. Taken together, the evidence reported in these studies indicates that an increase in glutamate tone is important for the establishment of irritable status in epilepsy.

Both NMDA and AMPA receptors are expressed in glial cells. Gliosis is a common pathological feature of the epileptic brain [145], and it is possible that reactive astrocytes may have a different function to non-reactive astrocytes. The expression of glutamate receptors, particularly AMPA receptors, is not uniform across normal brain astrocytes [146]; however, it has been shown that astrocytes derived from epilepsy patients do express functional AMPA receptors [147]. Evidence demonstrates that activation of AMPA receptors in glial cells causes rapid inhibition of inward-rectifier potassium (Kir) channels [148], and in Bergmann glial cells results in release of glutamate as a gliotransmitter [5]. On the other hand, expression levels of NMDA receptors on astrocyte membranes seemed to be lower compared with other glutamate receptors, and their functional significance in this setting is not well understood [146]. However, prolonged exposure of astrocytes to NMDA results in decreased expression of the functional astrocyte-specific proteins glutamine synthetase and the aquaporin-4 water channel, and also reduces the expression of Kir4.1 [149,150]. The results described above indicate that AMPA and NMDA receptors may play a role in altering astrocyte status, leading to enhanced brain excitability. Furthermore, AMPA-receptor–expressing astrocytes showed a loss of intercellular connection via GAP junctions [151]. GAP junctions are composed of two connexin or pannexin hemichannels from apposing cells, which dock to form a connecting channel that is closed to the extracellular space. GAP junction uncoupling therefore results in the opening of hemichannels, allowing efflux of ATP, glutamate, and potassium ions, which could lead to destabilization of the membrane potential and enhance inflammation and excitability in the brain [152]. Both NMDA and AMPA receptors may therefore contribute to expression and maintenance of abnormal excitability at the seizure focus, and under these conditions of elevated glutamate levels, AMPA and NMDA receptor activation could potentiate unstable neuronal activity, leading to hyperexcitation. This status may represent a target condition for glutamate antagonists.

### 5.2. Hyperexcitability

As noted previously, an elevation in glutamate levels and high-frequency oscillations are observed prior to the initiation of an ictal event [10,137,138,139,140]. Similar pre-ictal discharge was observed in brain slice specimens from patients with epilepsy. The AMPA receptor antagonist NBQX, but not the NMDA receptor antagonist AP-5, inhibited pre-ictal-discharge–like activity [153], while the AMPA/kainate receptor antagonist CNQX also reduced the number of occurrences of intra-hippocampal fast ripples in a rat model of kainate-induced spontaneous seizures [154]. This inhibition of discharge by AMPA antagonists supports a role for glutamate spillover in pre-ictal hyperexcitability, and suggests a greater contribution of AMPA receptors than NMDA receptors to hyperexcitability.

### 5.3. Synchronized Activity

A paroxysmal depolarization shift (PDS) is the experimentally measurable manifestation of the hypersynchronized synaptic transmission underlying ictal events. NMDA receptor antagonists have been shown to shorten the duration of PDS, whereas AMPA receptor antagonists completely suppress PDS evolution, indicating the importance of initial synchronization of synaptic transmission in PDS generation [155], given that rapid depolarization is mediated by AMPA receptor, while NMDA receptor activation occurs only after the elevation of membrane potential, and is prolonged due to slow desensitization.

Environmental stimulation (high potassium and low magnesium) induced ictal events in brain slices from patients with epilepsy [153], which appeared to be mediated by weak NMDA receptor activation. However, antagonists of both AMPA and NMDA receptors could inhibit the emergence of ictal events, with results suggesting that both types of ionotropic glutamate receptor, and especially the AMPA receptor, contribute to the generation of synchronized events. Considering the available evidence, it appears that AMPA receptors have a larger contribution than NMDA receptors to the initiation of PDS and ictal events.

Ictal events are relatively easily elicited in an experimental setting. In general, experimental animal models exposed to electrical stimulation become drug-refractory with increased stimulus intensity. For example, the amygdala kindling model, which is often utilized in the evaluation of anti-seizure drugs, becomes refractory to sodium channel blockers, levetiracetam, and valproate with increased stimulus intensity. On the other hand, the AMPA receptor antagonist perampanel has been shown to shorten the after-discharge duration and prolong latency to onset of generalized seizures [156]. This shortening of the after-discharge duration at the seizure focus indicates that AMPA receptor antagonists can reduce neuronal synchronized activity.

Animal models of status epilepticus show self-sustained synchronized activity. Several studies have reported that AMPA receptor antagonists can terminate status epilepticus in animal models [157,158,159,160,161,162,163,164]. A recent study demonstrated that perampanel terminated status epilepticus in a pilocarpine model of status epilepticus, but amantadine, an NMDA receptor antagonist, did not [163]. Similarly, the NMDA antagonist ketamine did not demonstrate a stable effect in animal models of status epilepticus when administered as monotherapy, but showed synergistic efficacy when administered in combination therapy with other drugs [165,166,167,168]. As such, AMPA antagonists appear to have an advantage over NMDA antagonists in the termination of self-sustained synchronized activity. These results also suggest that AMPA receptors have a more significant role in synchronized neuronal activity than NMDA receptors.

Somatosensory evoked potentials (SEPs) can be identified by EEG on the basis of their components, which are named according to their polarity (i.e., positive (P) or negative (N)) and typical peak latency (e.g., N20, P25, N33). SEPs with an amplitude of >10 µV between N20 and P25 are called giant SEPs (gSEPs), which are often observed in cortical myoclonus [169], including patients with progressive myoclonic seizures [170]. gSEPs are considered to be enlarged synaptic potentials with enhanced synchronism, and are therefore similar to PDS. Oi et al. [118] evaluated the effect of perampanel on gSEPs, and found that perampanel reduced gSEP amplitude. Interestingly, perampanel efficacy related to prolonged latency to the P25 and N33 components of gSEP. These results could be explained by temporal dispersion of synaptic activity. It is known that AMPA receptor activity mediates fast synaptic transmission and has an important role in the early phase of excitatory synaptic potential, whereas NMDA receptors have a role in delayed and prolonged synaptic transmission; this temporal dispersion effect may be an important factor when considering the inhibition of synchronized activity in ictal discharge (Figure 3). AMPA receptor antagonists inhibit the early phase of synchronization (ictogenesis), which may explain why perampanel has demonstrated broad-spectrum anti-seizure effects in clinical studies. The apparent lack of, or weak, efficacy of NMDA antagonists might be explained by their weak effects against synchronized activity and enhancement of neuronal excitation through an elevation in glutamate concentration via inhibition of inhibitory neurons.

## 6. Summary/Conclusions

Irregular activity of both NMDA and AMPA receptors may be a cause of seizure disorders. The pathology of irregular NMDA receptor function appears to include two different endophenotypes: hypo-NMDA and hyper-NMDA function. Hyper-NMDA–type epilepsy tends to show continuous EEG change. In the past, NMDA antagonists were only evaluated in patients with focal seizures, and clinical study results have shown that the NMDA receptor antagonist D-CPP-ene causes aggravation of focal seizures, suggesting that NMDA receptor antagonists may be unsuitable for the treatment of this seizure type. It is possible that NMDA receptor antagonists may be suitable in conditions of hyper-NMDA function; however, clinical phenotypes produced by gain-of-function mutations have been variable, making it difficult to establish seizure types characterized by hyper-NMDA function. In addition, the NMDA receptor seems less important for generation of synchronized activity than the AMPA receptor, which may be reflected in reports of weaker efficacy of NMDA receptors when administered as monotherapy, compared with as adjunctive therapy. On the other hand, it appears that AMPA receptors have a more prominent role in seizure disorders. An increase in surface expression of AMPA receptors could be elicited by various conditions, including genetic mutations affecting the receptor-trafficking system or following an event of synaptic plasticity. In addition, AMPA antagonists have been reported to exert effects on various stages of ictogenesis. In particular, the reduction of hyper-synchronization appears to be a specific advantageous feature of AMPA antagonists. It is possible that this underlies the broad-spectrum efficacy of the AMPA antagonist perampanel in various seizure types.

Knowledge of the precise pathophysiology of epilepsy, and how it is influenced by pharmacological interventions, may enhance awareness of the disease and support further anti-seizure drug development. Currently, AMPA and NMDA receptors represent promising targets for drug development in epilepsy and other diseases. An increased understanding of the roles of these glutamate receptors in the pathophysiology of epilepsy may therefore aid optimal utilization of clinically available drugs, as well as the identification of potential new drug candidates in the development pipeline.

## Figures and Tables

**Figure 1 biomolecules-10-00464-f001:**
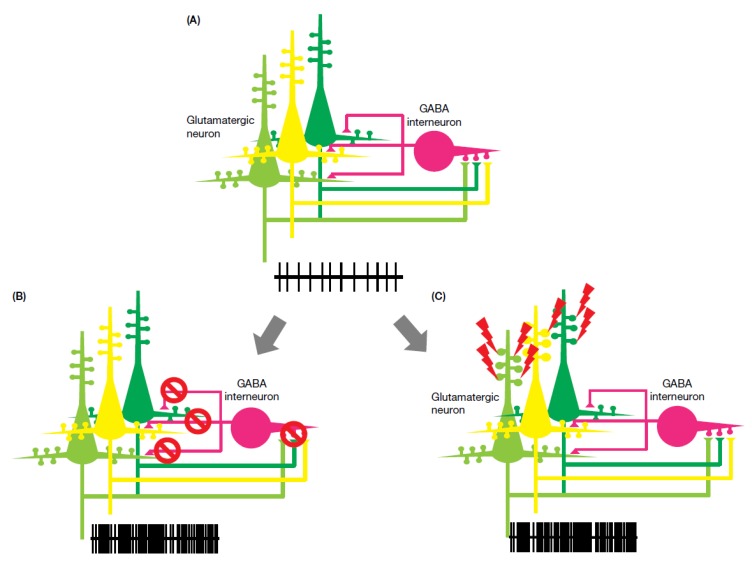
Physiological and pathological N-methyl-D-aspartate (NMDA) receptor function. (**A**) Physiological interaction between excitatory and inhibitory neurons. (**B**) Hypo-NMDA function: excitatory input to the inhibitory neuron is diminished by hypo-function of the NMDA receptor; the silencing of an inhibitory neuron results in an increase in excitatory neuron firing. (**C**) Hyper-NMDA function: a gain-of-function mutation could enhance neuronal excitation. NMDA, N-methyl-D-aspartate; GABA, gamma aminobutyric acid.

**Figure 2 biomolecules-10-00464-f002:**
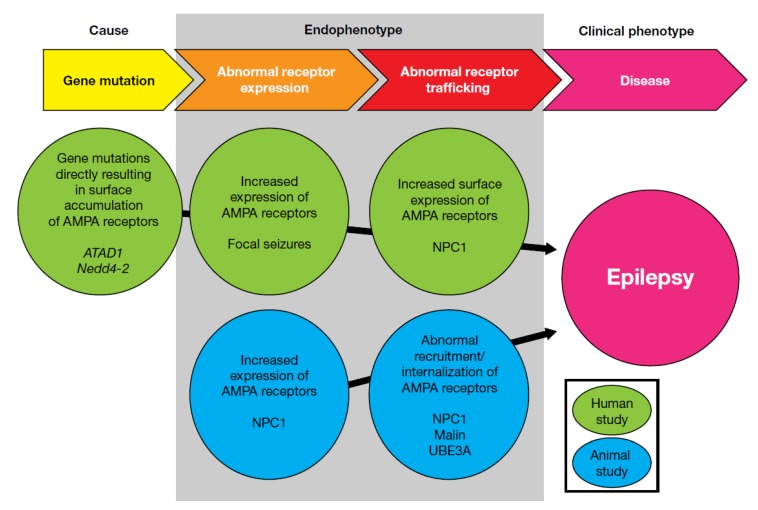
Summary of evidence for AMPA-receptor–related endophenotypes underlying clinical epilepsy. AMPA, α-amino-3-hydroxy-5-methylisoxazole-4-propionic acid.

**Figure 3 biomolecules-10-00464-f003:**
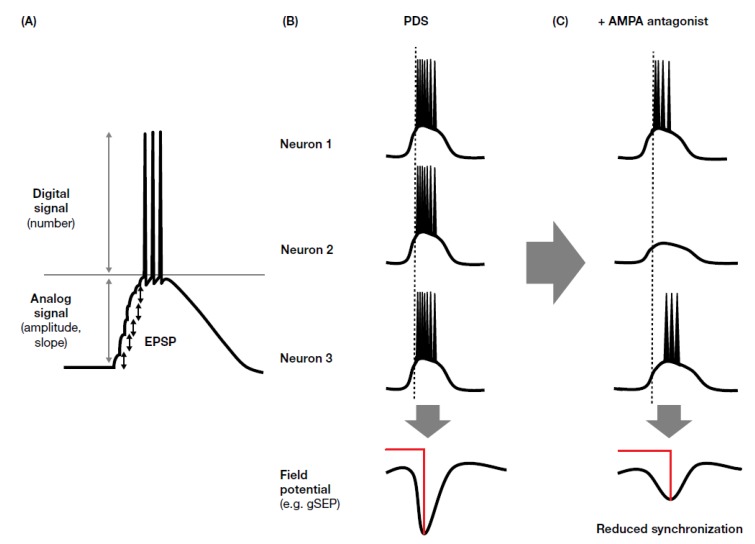
Inhibitory effect of AMPA antagonists on PDS. (**A**) Synchronized synaptic activity is the origin of PDS. AMPA-receptor–mediated EPSP accumulation occurs in the early phase of a PDS. (**B**) Concurrent accumulation of PDS elicits a gSEP. (**C**) AMPA antagonists inhibit the early phase of a PDS, resulting in temporal dispersion and the suppression of synchronized activity. AMPA, α-amino-3-hydroxy-5-methylisoxazole-4-propionic acid; EPSP, excitatory postsynaptic potential; gSEP, giant somatosensory evoked potential; PDS, paroxysmal depolarization shift.

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
