# Peer review of "Ionotropic Glutamate Receptors in Epilepsy: A Review Focusing on AMPA and NMDA Receptors"

_biomolecules, 2020, doi:10.3390/biom10030464_

Round 1

Reviewer 1 Report

This is a well written review with a detailed discussion of recent findings and good coverage of new and classic literature in the field. The author comes up with new ideas of using iGluR antagonist in the treatment of epilepsy that differentiate between the contribution of NMDA versus AMPA receptors.

Line 46: It would be helpful to explain the term ictal and other related terms to those readers who are not familiar with the topic.

Fig. 1: it appears that both NMDA hypo- and hyper-function result in increased excitability.  How does the system experience reduced excitability with increased inhibition?

Line 239: what is 'AAA+ ATPase Thorase', what is 'AAA+'?

Line 400-7: this background information should come much earlier, namely, when the author first talks about ictal activity.

Line 434: a verb  is missing 'The gSEPs (are?) often observed in patients with cortical myoclonus...'

Line 437: what are the 'P25 and N33 components of gSEP'?

Reviewer 2 Report

In this review article, Hanada discussed the role of AMPA and NMDA glutamate receptors in epilepsies.  The topic is interesting, and the Summary/Conclusion is reasonable based on the literature and clinical studies.  The manuscript is well organized and written.  My comments are as follows:

  • Readers will appreciate it if a section on future research direction is included after the Summary/Conclusion section.  
  • In the third paragraph on page 7, the sentences “The impairment of receptor trafficking…compared with slices without interictal-spike-like activity” are complicated.  Please rewrite them as separate sentences.

Reviewer 3 Report

The manuscript by Hanada reviews the involvement of ionotropic receptors in epilepsy with special emphasis on AMPA and NMDA receptors. This is a very well written and informative paper. However, I miss discussing the role of AMPA and NMDA receptors (including the glycine site) in animal models of seizures and epilepsy. The works from the Loscher's group comes to mind (e.g. PMID: 9863655, PMID: 7982045, PMID: 8088343.).
